# DVOLVER: EFFICIENT PARETO-OPTIMAL NEURAL NETWORK ARCHITECTURE SEARCH

## ABSTRACT

Automatic search of neural network architectures is a standing research topic. In addition to the fact that it presents a faster alternative to hand-designed architectures, it can improve their efficiency and for instance generate Convolutional Neural Networks (CNN) adapted for mobile devices. In this paper, we present a multi-objective neural architecture search method to find a family of CNN models with the best accuracy and computational resources tradeoffs, in a search space inspired by the state-of-the-art findings in neural search. Our work, called *Dvolver*, evolves a population of architectures and iteratively improves an approximation of the optimal Pareto front. Applying *Dvolver* on the model accuracy and on the number of floating points operations as objective functions, we are able to find, in only 2.5 days[1], a set of competitive mobile models on ImageNet. Amongst these models one architecture has the same Top-1 accuracy on ImageNet as NASNet-A mobile with $8\%$ less floating point operations and another one has a Top-1 accuracy of $75.28\%$ on ImageNet exceeding by $0.28\%$ the best MobileNetV2 model for the same computational resources. [2]

## 1 INTRODUCTION

The advent of deep neural networks ushered in qualitative leaps in several challenging machine learning tasks including image recognition (Krizhevsky et al., 2017) and language modeling (Sutskever et al., 2014). However, on top of being tedious and time consuming, deep neural network design requires deep technical knowledge. Automating this design process has reached important milestones in the last few years as advances in neural architecture search (Zoph & Le, 2016; Zoph et al., 2017) bridged the performance gap between manually designed architectures and those found via automatic search.

Traditional approaches usually rely on Reinforcement Learning (RL) (Baker et al., 2016; Zoph et al., 2017; Zoph & Le, 2016; Zhong et al., 2017), Genetic Algorithms (GA) (Precup & Teh, 2017; Xie & Yuille, 2017; Liu et al., 2017b; Real et al., 2018) or Sequential Model Based Optimization (SMBO) (Liu et al., 2017a; Brock et al., 2017), to search for the best performing architectures.

This search process is quite resource heavy which is why current work on architecture search shifted focus from merely outperforming manually designed architectures to accelerating the search process (Liu et al., 2017a; Pham et al., 2018; Liu et al., 2018). Nonetheless, very few works (Dong et al., 2018; Smithson et al., 2016; Ye-Hoon et al., 2017) attempted to take additional constraints such as memory consumption or inference time into consideration when evaluating candidate architectures, although it is of crucial importance when targeting mobile devices.

In this article, we introduce *Dvolver*, a principled multi-objective neural architecture search approach using Pareto optimality to navigate the search space. We instantiate this approach with a standard evolutionary algorithm (Deb et al., 2002) on the NASNet search space (Zoph et al., 2017) with validation accuracy and the total number of floating point operations (FLOPS) as objectives.

Our experiments show that (i) we do not sacrifice accuracy by looking for faster architectures (ii) we are able to explore the search space without loss in efficiency (iii) we are able to find architec-

---

[1] Using 20 NVIDIA V100 GPUs
[2] Code and checkpoints are publicly available at `https://goo.gl/1QZX6a`.

tures that outperform NASNet-A mobile and certain configurations of MobileNetV2 architectures (Sandler et al., 2018) when accounting for both accuracy and floating point operations count.

## 2 RELATED WORK

Our work is founded on previous successful contributions focusing on single objective neural network architecture search that we extend for multiple criteria. Recently, independent papers came out with a similar approach but their search space is orders of magnitude smaller than the best studies so far.

**Mono-objective Search**   Zoph et al. (2017) introduce a modular search space, based on reusable cells, which not only reduces the computational cost of the search process but also allows the search to be carried out on a small database and transfered to a larger one. They use RL to train a Recurrent Neural Network (RNN) that generates the structure of the cells. Real et al. (2018) showed that artificially-evolved architectures, on the same search space, are at least as good as their RL counterpart. On a similar note, Liu et al. (2017a) adopted SMBO coupled with a progressive exploration of a variation of said space, greatly reducing the search cost while maintaining competitive results. Their approaches achieved state of the art performances on ImageNet (Deng et al., 2009), with modules learned on CIFAR-10 (Krizhevsky & Hinton, 2009) but they only optimized for best validation accuracy. The best module is then combined in a scaled down network and compared with the best mobile architecture at the time. Even though, the results are impressive, this is sub-optimal and explicitly optimizing for validation accuracy and speed can lead to even better architecture.

**Multi-objective Search**   Smithson et al. (2016) used Pareto optimality as a selection criteria in an SMBO-based hyperparameter search, and showed that the obtained Pareto front is close to the one obtained via exhaustive search, even with two orders of magnitude fewer architecture evaluations. However, their work focused on hyperparameter search and did not explore a more general search space.

Elsken et al. (2018) used network morphism to optimize network architecture search on multiple criteria namely test error and parameters count. While an interesting approach, this paper lacks clear comparison with single objective methods and only evaluates architecture transferability on ImageNet64x64 (Chrabaszcz et al., 2017).

In Ye-Hoon et al. (2017), a multi-objective genetic algorithm is implemented to search for fast and accurate architectures, initializing the search process from a given baseline network. They were able to find architectures that outperformed common baselines [3] both in speed and accuracy on MNIST and CIFAR10. Their method, while in principle similar to ours, explored the vicinity of a defined network and was not extended to a complex search space where the search process is initialized arbitrarily. Furthermore, it was not validated for learning architectures that transfer to large scale applications.

Dong et al. (2018) work is perhaps the closest to ours in intent and scale, as it performs multiple objective architecture search using Pareto optimality and uses a proxy dataset to avoid an expensive search on ImageNet. It even goes one step further to incorporate device-dependent metrics. However, their search space is orders of magnitude smaller than ours in addition to being heavily biased towards mobile architectures.

## 3 PROPOSED METHOD

For a given search space, our goal is to find all the neural network architectures that offer the best trade-offs between multiple independent criteria. We formulate the search process as a *multi-objective optimization* (MOO) problem (Section 3.1) where each objective to optimize is an independent scalar value. We refer to Figure 1 for a general overview of the search process.

The MOO controller selects child networks with different architectures. The child networks are evaluated on multiple criteria by the Evaluation Engine. The resulting objective values are then used

---

[3] namely Lenet (Lecun et al., 1998) and an all convolutional net (Springenberg et al., 2014)

by the controller as feedback so it can generates better architectures over time. At each iteration, *Dvolver* keeps the hall of fame of the best architectures found so far.

## 3.1 MULTI-OBJECTIVE OPTIMIZATION AND PARETO DOMINANCE

We define the *search space* $X$ as the set of all feasible solutions and the *objective space* as the set of all possible values of the objective function $f : X \rightarrow \mathbb{R}^k$. *Dvolver* searches for solutions by maximizing the objective function $f$ over $X$ with the total order relation defined next.

For a nontrivial multi-objective optimization problem, there is no solution that simultaneously optimizes all the objectives. In that case, the objective functions are said to be *conflicting*, and there exists a (possibly infinite) number of compromise solutions called *Pareto-optimal* solutions. The set of Pareto optimal outcomes is often called the *Pareto front*.

A simple approach to cope with multi-objective functions is to use scalarization which squashes all the objective functions into a single function falling back to single-objective optimization. While very simple in practice, this method suffers from many shortcomings such as (i) the

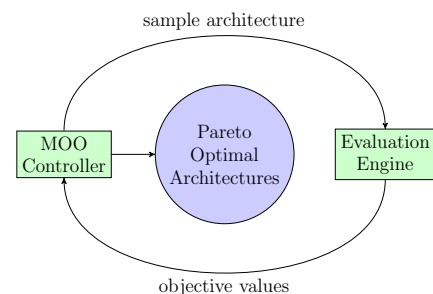

sample architecture

objective values

Figure 1: Overview of *Dvolver* Search Process. A controller generates a candidate architecture which is trained to convergence by the evaluation engine. Other criteria to optimize are evaluated while the network is training. Resulting metrics are fed back to the controller. At each iteration, the controller keeps track of the best performing architectures.

difficulty to combine multiple unrelated criteria into a single value and (ii) inefficient sampling of the full Pareto front.

A much better solution is to use Pareto dominance partially ordered relation defined as follow: A solution $v$ Pareto-dominates a solution $u$, if all objective values for $v$ are greater than the objective values for $u$ but a least one is strictly superior. A feasible solution is Pareto optimal, if it is not dominated by another solution. Without additional subjective preference information, all Pareto optimal solutions are considered equally good.

For the search process to be effective, it needs a totally ordered relation. Deb et al. (2002) defines the *crowding distance* value of a particular solution $d_i$ of point i as a measure of the objective space around i which is not occupied by any other solution in the population. Here, we simply calculate this quantity $d_i$ by estimating the perimeter of the cuboid (Figure 8) formed by using the nearest neighbors in the objective space as the vertices (See appendix A.2 for more details). In addition to the Pareto dominance criteria, the crowding distance defines the totally ordered relation used by *Dvolver* to perform its search process.

## 3.2 SEARCH ALGORITHM

*Dvolver* evolves a population of $N$ individual architectures. At each iteration, the population evolves closer and closer toward the Pareto front, effectively sampling the full Pareto front in a single search process.

Our method starts by generating a random population and for each generation, it generates a child population, evaluates each child by training the networks for a few epochs on CIFAR-10 and selects the $N$ dominant architectures from the $2 \times N$ individuals (parents and children). The details of the algorithm is presented in appendix A.1. Algorithm convergence is monitored with the hypervolume indicator defined as the volume under the Pareto front (see A.3). In Algorithm 1, we present the *Dvolver* selection process inspired by NSGA-II (Deb et al., 2002).

At the end of an iteration, there are $N$ parents and $N$ children. We should select the best $N$ individuals to form the parents' population for the next iteration. We compute the Pareto fronts for the $2 \times N$ individuals. The best fronts are considered first. If there are more than $N$ individuals in the

---

**Algorithm 1:** *Dvolver* selection algorithm

---

    **Inputs**
- $N$ individuals from parents' population
- $N$ individuals from offsprings' population

    **Output**
- $N$ selected individuals $\text{Inds}_{sel}$

1: $\text{Inds}_{sel} \leftarrow \varnothing$;
2: $\text{fronts} \leftarrow \texttt{ComputeParetoFronts}(\text{parents} \cup \text{offsprings})$;
3: $i \leftarrow 0$;                                                    ▷ Starts with Pareto front with rank 0
4: $\text{rem} \leftarrow N$;                     ▷ rem is the number of remaining individuals to be selected
5: **while** *rem* > 0 **do**
      $\text{Inds}_i \leftarrow \texttt{SortWithDominance}(\text{fronts}_i)$;
      **if** $\texttt{Len}(Inds_i) < rem$ **then**
          $\text{Inds}_{sel} \leftarrow \text{Inds}_{sel} \cup \text{Inds}_i$;
          $\text{rem} \leftarrow \text{rem} - \texttt{Len}(\text{Inds}_i)$;
          $i \leftarrow i + 1$;                       ▷ Move to the next Pareto front
      **else**
          $\text{Inds}_{sel} \leftarrow \text{Inds}_{sel} \cup \texttt{First}(\text{rem}, \text{Inds}_i)$;  ▷ Highest crowding distance individuals
      first rem $\leftarrow 0$;

---

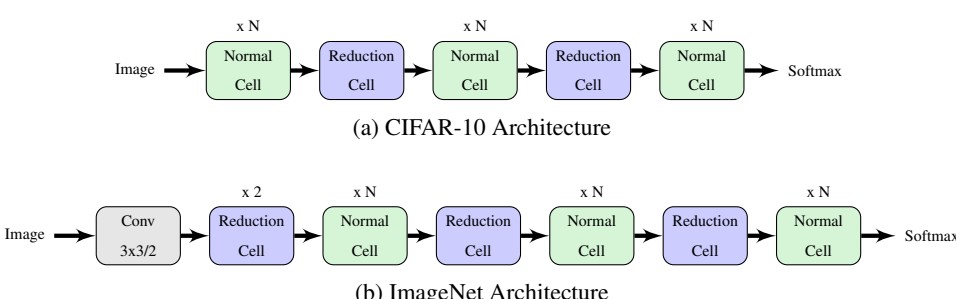

Figure 2: Scalable architectures for image classification consist of two repeated motifs termed Normal Cell and Reduction Cell. This diagram highlights the model architecture for CIFAR-10 and ImageNet. The choice for the number of times the Normal Cells that gets stacked between Reduction Cells, N, can vary in our experiments.

best front, the first $N$ (with the help of the crowding distance) are selected and the selection ends. Otherwise, all the individuals from the front are selected and we carry on with the next best Pareto front until $N$ individuals are selected.

### 3.3 SEARCH SPACE

Our work is based on the work of Zoph et al. (2017), in which network architecture is composed of two levels: a micro level architecture defined by parametrized cells composed of $B$ blocks and a macro architecture where those cells are stacked a predefined number of times, in order to create the final network. Figure 2 describes the macro architecture used for CIFAR-10 and ImageNet architectures.

There are two types of cells and *Dvolver* searches for both at the same time: (i) *Normal Cell* that returns a feature map of the same size as its input and (ii) *Reduction Cell* that returns a feature map of half the size of its input in the spatial dimensions.

A cell is composed of $B$ *blocks* arranged in a Directed Acyclic Graph (DAG). Each block is a mapping of two input tensors to one output tensor and it can be represented by a 4-tuple, $(I_1, O_1, I_2, O_2)$ where $I_k$ specifies the input for the operation $O_k$. The result of the two operations $O_1$ and $O_2$ are summed together to form the output of the block. A block can take its input from previous blocks,

the previous cell $c_{k-1}$ or the previous previous cell $c_{k-2}$. Unused blocks are concatenated in the depth dimension to form the final output of the cell.

Moreover, we noticed that the original NASNet search space can significantly benefit from extra connections from any given block, the previous cell and the previous previous cell to the final concatenation layer. In fact, these extra connections are already present but not described in the best models from Zoph et al. (2017), Real et al. (2018) and Liu et al. (2017a). Instead of adding manually these connections after the search process like in the previous works, *Dvolver* optimizes for these extra connections in addition to the 4 parameters for each block. In practice, our search space is the same as the one used by Zoph et al. (2017) but no manual intervention is needed. For $B = 5$, the total number of parameters for each type of cell is 20 parameters for the blocks and one list of indexes of extra connections.

Similar to the approach of Liu et al. (2017a), we simplify the search space and consider only the following operations:

- identity
- 3x3 average pooling
- 3x3 max pooling

- 3x3 depthwise-separable conv
- 5x5 depthwise-separable conv
- 7x7 depthwise-separable conv

The total number of combinations for the defined search space is about $10^{20}$ taking into account symmetries and possible redundant connections. Note that our search space is larger than the one defined in Liu et al. (2017a) but we show in Section 4 that *Dvolver* is capable of finding competitive architectures faster.

## 4    EXPERIMENTS AND RESULTS

Here we present the results for *Dvolver* with the method described in Section 3.1 and search space in Section 3.3. Searching for neural network architecture is done on CIFAR-10 (see Section 4.1) and optimization is performed with the validation accuracy and the number of floating points operations as independent objectives. In fact, *Dvolver* works by maximizing its objective functions, so in practice, the first objective function is the validation accuracy and the second is the speed $= \frac{2 \times 10^9}{\text{FLOPS}}$. The best architectures are then fully trained on ImageNet (see Section 4.3).

### 4.1    SEARCH RESULTS

In this section, we describe our experiment to learn the Pareto front of convolutional cells for the search space described above. In particular, we conduct the search process on the CIFAR-10 dataset. We separate 5,000 images from the train set as a validation set used for architectures evaluation during the search process. We use a population of 32 individuals and a uniform mutation and a crossover probability of 0.1. Our preliminary studies shows that doubling or dividing by two these parameters does not have a large impact on the search process. We chose uniform operators instead of 1-point or 2-points operators to favor small architecture modifications: the different architectures parameters are modified independently of the other parameters.

The number of cells $N$ is set to 2 and use 32 convolution filters. Each sampled architecture is trained for 72 epochs with a batch size of 150 and a learning rate with a linear decay from 0.1 to 0.039. We evaluate the candidate architectures in parallel on multiple GPUs and the search process takes 50 GPU-days to find competitive architectures.

The Figure 3 shows the evolution of the hypervolume indicator as a function of the sampled architectures (Figure 3(a)) and the final Pareto front estimation with 2 intermediate fronts as an illustration of the evolution process. The horizontal segment in the Pareto front are particularly interesting. *Cells situated on knees in the Pareto front are good trade-offs: for a small drop in accuracy there is a large gain in speed.* Figure 3 shows how the Pareto front can be an appropriate tool when choosing a network architecture for a specific application.

Training all the Pareto optimal solutions in the front on ImageNet is very expensive and of limited interest in practice. Usually, when searching for efficient architecture for mobile, one is interested

on an architecture with specific computational resources: it can be inference time or in our case the number of floating point operations. We manually select 3 architectures situated on the Pareto front knees that will be fully train on CIFAR-10 in Section 4.2 and on ImageNet in Section 4.3. They are called Dvolver-A, B and C. We choose $N$ and $F$ so that final neural networks have similar floating point budget to other neural networks in the competition for easy comparison.

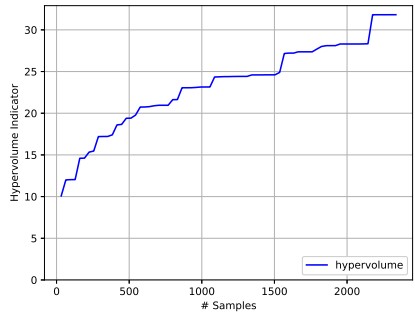

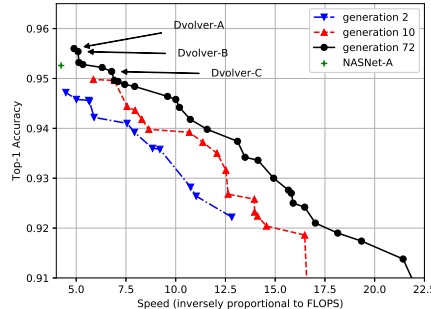

(a) Hypervolume indicator evolution

(b) Evolution of the Pareto front estimation

Figure 3: 3(a) shows the evolution of the hypervolume during the search process until it stops increasing. 3(b) represents the Pareto front estimation for a few generations during the search process. We define the speed $= \frac{2 \times 10^9}{\text{FLOPS}}$ which can be roughly interpreted as the number of inferences per second on a mid-range mobile device. The 3 selected cells Dvolver-A, B and C are shown on the Pareto front and NASNet-A is represented for reference.

The detailed architectures of the cells for Dvolver-A, B and C are presented in the diagrams of Figure 4. We can see that, in most of them, additional connections (see Section 3.3) are present. This matches our observations during our preliminar experiments. On the contrary to previous works (Zoph et al., 2017), (Real et al., 2018) and (Liu et al., 2017a) where additional connections are added manually after the search process, our method is able to find them automatically.

## 4.2 RESULTS ON CIFAR-10

We first present the results for the 3 architectures selected in section 4.1. We use the same pre-processing strategy as Zoph et al. (2017) namely the 32×32 images are pad with zeros to 40×40 images, then we select a random crop of 32×32, apply random horizontal flip, perform random brightness and contrast variations and finally apply cutout preprocessing (Devries & Taylor, 2017).

First, we compare our method to single-objective methods taking into account only the test accuracy (see top half of Table 1). Parameters count is presented for reference. *Dvolver* is able to find network architectures with better test error than NASNet-A.

Then, we compare with multi-objectives methods (see bottom half of Table 1). While the search spaces are very different, *Dvolver* finds architectures with better test error than Dong et al. (2018) and is on part and even surpass the networks proposed by Elsken et al. (2018).

## 4.3 RESULTS ON IMAGENET

Dvolver-A, B and C cells found in Section 4.1 are fully trained for different combination of cells, number of normal cells $N$ and number of convolution cells $F$ on ImageNet following the training procedure found in Real et al. (2018). Input image size is set to 224x224 and each network is trained for 330 epochs. We use the standard inception preprocessing, train with rmsprop optimizer and exponential learning rate decay with initial learning rate of 2, 2.4 epochs per decay with 0.97 learning rate decay value and warmup for 3 epochs. The total batch size is 1600.

First, in Table 2, we select a few architectures with similar computational complexity as networks presented in Zoph et al. (2017) and show that we can find Pareto dominant architectures that can be more accurate and faster than architectures found with single objective optimization. Dvolver-A (N=4, F=44) has the same Top-1 accuracy of $73.64\%$ with $8\%$ less floating point operations than

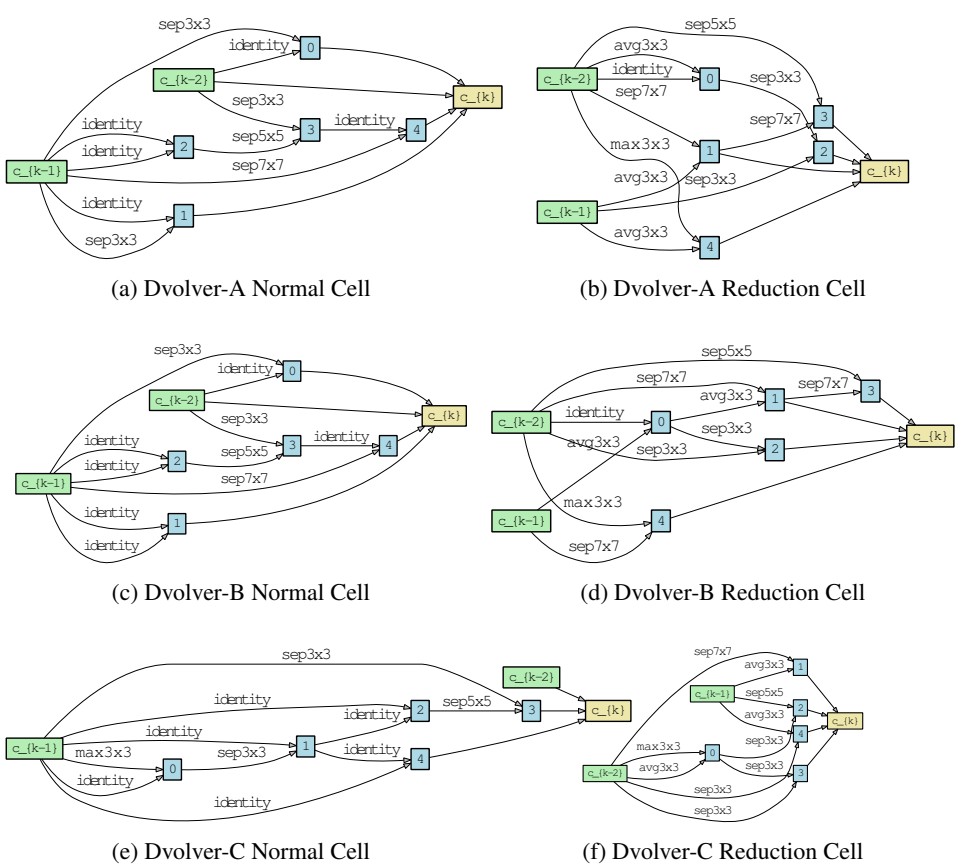

Figure 4: Dvolver-A, B and C cells (best viewed in color). Blue boxes are the output of the 5 blocks. Each block has 2 incoming arrows representing the 2 operations in the block. Labels on arrows describe the specific operation performed. The origin of the arrow is the input of for the operation. Green boxes represent the previous and previous previous cells and the yellow box represents the final depth concatenation at the end of the cell.

Table 1: Performance of Dvolver and competitive methods on CIFAR-10. NASNet results are the ones we could reproduce in our setup on the same training conditions as the rest of our results. There are 2 FLOPS for 1 Mult-Adds (MACs).

| Model | Top-1 Err.(%) | #Mult-Adds | #Parameters |
|---|---|---|---|
| NASNet-A (N=6, F=32) | 2.93 | 545M | 3.3M |
| Dvolver-A (N=5, F=36) | 2.9 | 505M | 3.17M |
| Dvolver-B (N=6, F=32) | 3.03 | 436M | 2.76M |
| Dvolver-C (N=6, F=32) | 3.04 | 315M | 2.08M |
| DPP-Net-PNAS | 4.36 | 1364M | 11.39M |
| DPP-Net-WS | 4.78 | 137M | 1.00M |
| DPP-Net-ES | 4.93 | 270M | 2.04M |
| DPP-Net-M | 5.84 | 59.27M | 0.45M |
| DPP-Net-Panacea | 4.62 | 63.5M | 0.52M |
| Lemonade Cell9 (V1) | 4.57 | - | 0.5M |
| Lemonade Cell9 (V2) | 3.69 | - | 1.1M |
| Lemonade Cell2 (V1) | 3.05 | - | 4.7M |
| Lemonade Cell2 (V2) | 2.58 | - | 13.1M |

NASNet-A mobile. We also found two architectures with slightly more computations needs (3% and 7% more MACs) but with better Top-1 accuracy 74.37% and 74.81% respectively.

Table 2: Performance of Dvolver and NASNet on ImageNet. NASNet results are the ones we could reproduce in our setup on the same training conditions as the rest of our results. It is on par with the median value found by Ramachandran et al. (2017) but lower than the official results (Top-1 74%) found by Zoph et al. (2017). There are 2 FLOPS for 1 Mult-Adds (MACs).

| Model | Top-1 Acc.(%) | Top-5 Acc.(%) | #Mult-Adds | #Parameters |
|---|---|---|---|---|
| NASNet-A (N=4, F=44) | 73.64 | 91.53 | 564M | 5.3M |
| Dvolver-A (N=3, F=48) | 73.14 | 91.37 | 493M | 4.3M |
| Dvolver-A (N=4, F=44) | 73.64 | 91.49 | **520M** | 4.5M |
| Dvolver-A (N=4, F=48) | **74.81** | 92.15 | 607M | 5.25M |
| Dvolver-B (N=3, F=52) | **73.65** | 91.72 | **544M** | 4.73M |
| Dvolver-C (N=3, F=44) | 70.24 | 89.58 | 317M | 2.76M |
| Dvolver-C (N=4, F=56) | **74.37** | 91.86 | 582M | 5.06M |

Inspired by the results of Ramachandran et al. (2017) on NASNet-A mobile, we replace all RELU activation functions by SWISH activation functions and compare our results with the best state-of-the-art architecture for mobile. We found that we can outperfom MobileNet-V2 in the high end mobile space with 75.28% Top-1 accuracy for the same number of floating point operations but stay behind on low-end mobiles. Table 3 summarizes our findings:

Table 3: Performance of Dvolver and other state-of-the-art models on ImageNet. Result for NASNet-A swish is the median found in Ramachandran et al. (2017).

| Model | Top-1 Acc.(%) | Top-5 Acc.(%) | #Mult-Adds | #Parameters |
|---|---|---|---|---|
| NASNet-A (4, 44, swish) | 74.9 | 92.4 | 564M | 5.3M |
| MobileNetV1-224 1.0 | 70.9 | 89.9 | 569M | 4.24M |
| MobileNetV1-224 0.75 | 68.4 | 88.2 | 317M | 2.59M |
| MobileNetV2-224 1.4 | 75.0 | 92.5 | 582M | 6.06M |
| MobileNetV2-224 1.3 | 74.4 | 92.1 | 509M | 5.34M |
| MobileNetV2-224 1.0 | 71.8 | 91.0 | 300M | 3.47M |
| MobileNetV2-224 0.75 | 69.8 | 89.6 | 209M | 2.61M |
| ShuffleNet (2x) | 70.9 | 89.8 | 524M | 5M |
| DPP-Net-Panacea | 74.02 | 91.79 | 523M | 4.8M |
| Dvolver-A (3, 48, swish) | 74.35 | 91.99 | 493M | 4.3M |
| Dvolver-B (3, 52, swish) | 74.70 | 92.33 | 544M | 4.73M |
| Dvolver-C (3, 44, swish) | 70.95 | 90.05 | 317M | 2.76M |
| Dvolver-C (4, 56, swish) | **75.28** | 92.32 | 582M | 5.06M |

There are a lot of different mobile devices with extreme computation resources diversity. To keep the discussion simple, we consider two tiers of devices: the low-end mobile market with devices with less than 400M Mult-Adds for a specific computation and a high-end tier with devices with higher performance.

Our best architectures are compared with the state-of-the-art architectures in Figure 5. In particular, we show that optimizing for multiple criteria including the number of floating operations, *Dvolver* finds architectures with better trade-offs than NASNet architectures. The poor performances at low operation count, show a weakness in the design of the NASNet search space which seems to be less efficient as a better tailored search space, design with low-end mobile device in mind.

## 4.4 SEARCH EFFICIENCY

An important result of our work is the computational resources needed to perform the architecture search. *Dvolver* finds the full Pareto front, on a much bigger search space, 3 times faster than PNAS (Liu et al., 2017a). Table 4 presents the computational resources needed to perform the automatic search process with different method found in the field.

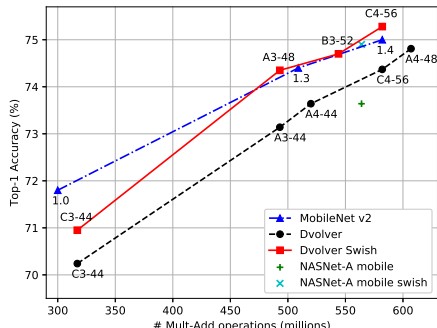 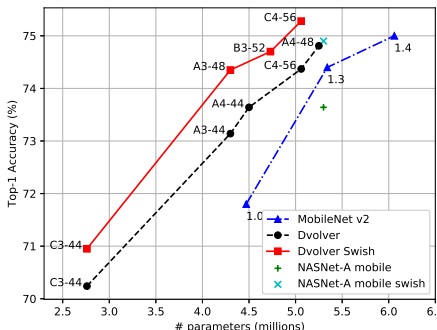

Figure 5: Accuracy versus computational demand (left) and number of parameters (right) for *Dvolver* and state-of-the-art architectures for mobile devices on ImageNet. Computational demand is measured in the number of floating-point multiply-add operations to process a single image. Labels on the figures for *Dvolver* are of the form Family, N and F. For example Dvolver-A with N=3 and F=48 is noted A3-48.

Table 4: Search efficiency for *Dvolver* and other state-of-the-art related methods. DPP-Net values are for 4 layers as described in Dong et al. (2018).

| Method | GPU.days | Search Space Size (Approx.) |
|---|---|---|
| NASNet (Zoph et al., 2017) | 1800 | $10^{28}$ |
| PNAS (Liu et al., 2017a) | 150 | $10^{12}$ |
| AmoebaNet (Real et al., 2018) | 3150 | $10^{28}$ |
| Dvolver | 50 | $10^{20}$ |
| DPP-Net | 8 | 324 |

One explication for *Dvolver* speed compared to other method is that it actually generates a population of architectures and among them some are very fast (see Figure 3(b)). Intriguingly, these very fast architectures do not penalize the search process for the most accurate architectures. We think that the interaction between the different kind of individuals in the population is an essential property responsible for its effectiveness.

## 5 CONCLUSION

In this work, we propose a general method to search for Pareto-optimal neural architectures optimized for multiple criteria. In particular, we propose two novel cell structures. The first one achieves the same Top-1 accuracy on ImageNet as NASNet-A mobile with $8\%$ less floating point operations and the second one outperforms the best MobileNetV2 model by $0.28\%$ in Top-1 accuracy ($75.28\%$) for the same computational resources.

Our key contribution is the iterative population-based approach which approximates the Pareto front in a single search process. Our search method is efficient in terms of number of sampled architectures and computational resources. It performs the full search for the Pareto front in 50 GPU.days, 3 times faster than PNAS (Liu et al., 2017a) on a larger search space.

Our work is based on a few assumptions that can be discussed. First, it relies on CIFAR-10 as proxy for ImageNet in the search process. This has several weaknesses: It is very hard to distinguish different architectures on this dataset as the differences between architecture's accuracies are very small and very noisy. In addtition, our method relies on the order to be conserved when transfering the architectures from CIFAR-10 to ImageNet and in practice it works but it is not always the case. Second, it is very expensive to fully train each candidate architecture from scratch even on CIFAR-10. To speedup the search process, we train for a few epochs with high learning rate and state that

the ordering for this few epochs is the same as the one if all the architectures were fully trained. This is not always the case and it can inder the performance of the search process.

As a future work, we could apply our method on a different search space inspired by the lastest state-of-the-art mobile architectures to improve the performance on low-end mobile devices. Another way to improve our work is to enhance the selection phase by reducing the variance in the accuracy caused by the early stopping assumption.

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

# A  ALGORITHM DETAILS

## A.1  GENETIC ALGORITHM

Figure 6 shows the details of the genetic algorithm used by *Dvolver*. The optimization process looks for a global optimum of the fitness function. It operates on a population of individuals (each being a point in the search space). Each iteration (also called generation) consists in selecting individuals from the parents' population, applying crossover (also known as recombination) and mutation operators to define a new offsprings' population. Each offspring's fitness function is then evaluated. In the end, the most fitted individuals are kept and become the parents' population for the next generation.

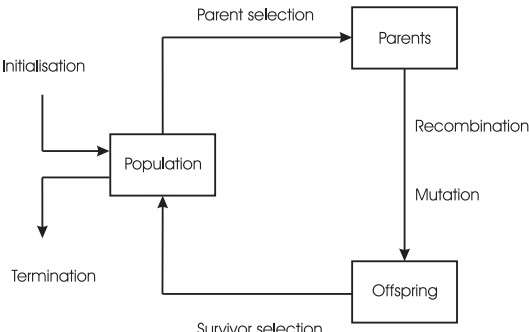

Figure 6: General Genetic Algorithm

Individuals are points in the search space and genetic operators modify their vectorial representation. These transformations generate new individuals that inherit some attributes from their parents but also acquire new ones. We choose uniform crossover operator (see Figure 7(a)) which evaluates each component in the parents' representation for exchange with a probability $\mu_{cross}$. It is particularly useful as it operates at the component level for fine-grained cell's architecture modifications. For the same reason, we choose uniform mutation operator (see Figure 7(b)) which replaces the value of each component with a probability $\mu_{mut}$, by a uniform random value selected between the possible choices for that component.

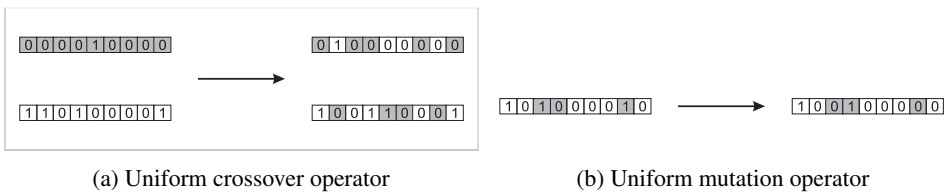

(a) Uniform crossover operator             (b) Uniform mutation operator

Figure 7: Crossover and mutation genetic operators

The exploration of new individuals in the search space versus the exploitation of the best candidates are governed by the two parameters $\mu_{cross}$ and $\mu_{mut}$, and are the unique two parameters for the genetic algorithm. Large value of $\mu_{cross}$ allows to take larger steps across the problem domain, allowing the escape of local optimum. $\mu_{mut}$'s value allows for small variations. It is possible to increase that value at the beginning of the search process to favor exploration then annealing that value as the population converge to a global optimum.

## A.2  CROWDING DISTANCE

During the search process, it is necessary to compare every couple of candidates and be able to say which one is the best. Pareto dominance is only a partially ordered relation and it is not able by itself to rank two solutions on the Pareto front. For that purpose Deb et al. (2002) introduces the crowding distance to favor isolated solutions on the front. The idea is that it should help sampling

the entire Pareto front. Pareto dominance and crowding distance defines a totally ordered relation that is independent of the ranges of the objective functions.

Figure 8 defines how the crowding distance is computed for each solution on the Pareto front. Red dots are part of the same Pareto front and the crowding distance for solution $i$ is the hypervolume of the cuboid. The crowding distance value of a solution provides an estimate of the density of solutions surrounding that solution.

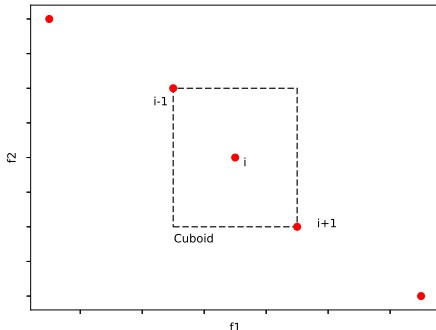

Figure 8: Crowding distance

### A.3 HYPERVOLUME INDICATOR

With Multi-objective Optimization, an algorithm produces a set of points in the objective space as an estimation of the Pareto front. A quantitative measure is desired to estimate the closeness of the estimated data points to the true Pareto front. One of such measures is the hypervolume indicator, which gives the hypervolume between the estimated Pareto front (P) and a reference point (R).

The problem of interest is the maximization of all the objective functions when all the objectives are positives. In that case, the reference point can be set to zero and the hypervolume indicator is the k-area under the Pareto front surface (see Figure 9). $hypervolume(n)$ is the hypervolume for the $n^{th}$ generations and is a monotonically increasing function with respect to the number of generations (or equivalently to the number of objective functions evaluations).

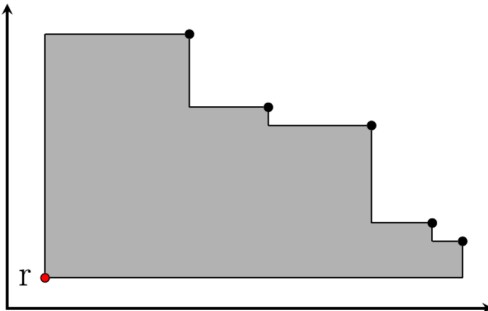

Figure 9: Hypervolume indicator

The hypervolume can be used to monitor the converge of the genetic algorithm: when the hyper-volume indicator stops increasing, we can consider that we have reached convergence. In practice, It may be necessary to wait a little longer in case the genetic algorithm has been stuck in local optimum.

