# OpenReview forum: "DVOLVER: Efficient Pareto-Optimal Neural Network Architecture Search"
_ICLR.cc/2019/Conference_

### Official Review · AnonReviewer2 · 2018-11-01
**Interesting but incremental work**

**Rating:** 4
**Confidence:** 4

**Review:**

The authors propose a multi-objective neural architecture search based on an evolutionary algorithm. The contradicting objective functions are optimized by ranking the candidates by Pareto-dominance, replace the bottom 50% with new candidates generated by the top 50% candidates through random mutations. The multi-objective function considers classification accuracy and an approximation of the inference speed. The method is compared to MobileNet and Mobile NASNet on ImageNet indicating an improvement with respect to search time.

The authors admit that their work is incremental and a combination of existing work. Furthermore, they admit that Dong et al. (2018) is the closest related work, however, they do not compare to them in the experimental section. The method by Dong et al. requires only 8 GPU days (Dvolver requires 50) yielding very similar results. Why this has been ignored remains unclear.

The paper is not self-contained, important methodological aspects of the method are insufficiently described. I recommend at least to formally define the crowding distance. It would be also reasonable to define your objective functions already in Section 3 instead of mentioning them in the caption of Figure 3 and its axis labels.

I think it's fair to call your approach evolutionary but you might want to discuss its relationship to beam search and in this scope discuss [A].

The comparison in Table 2 is not fair. You use the swift activation function and do not report the corresponding numbers for MobileNet or Mobile NASNet. Ramachandran et al. (2017) report these (75% and 74.2% for NASNet and MobileNet).
Comparing the Dvolver architecture with ReLU activations to MobileNet does not indicate any improvements.

You mention that most previous approaches are only keeping track of the best solution while you evolve over a population. Maybe this sentence is not well written and something else is meant but now this statement is wrong.

[A] Thomas Elsken, Jan Hendrik Metzen, Frank Hutter: Simple And Efficient Architecture Search for Convolutional Neural Networks. CoRR abs/1711.04528 (2017)

---

> ### Author Response · Authors · 2018-11-17
> **Arguments in favor of original work instead of just incremental**
>
> Thanks for your time and comments. Below we pull quotes from the review followed by responses.
>
> "The authors admit that their work is incremental and a combination of existing work. Furthermore, they admit that Dong et al. (2018) is the closest related work, however, they do not compare to them in the experimental section. The method by Dong et al. requires only 8 GPU days (Dvolver requires 50) yielding very similar results. Why this has been ignored remains unclear."
>
> Response: Thanks for pointing the missing comparison with DPP-Net, we updated the paper with their results.
>           We argue that the search space of DPP-Net is too small to prove that their method is efficient. There are only 324 possible architectures with 4 layers which can be searched exhaustively.
>           DPP-Net is an extension of Progressive NAS with multi-objective optimization but their search space is different. We cannot conclude on their effectiveness in these conditions.
>
>
> "The paper is not self-contained, important methodological aspects of the method are insufficiently described. I recommend at least to formally define the crowding distance. It would be also reasonable to define your objective functions already in Section 3 instead of mentioning them in the caption of Figure 3 and its axis labels."
>
> Response: Thanks for pointing the missing information. We update the paper with more details and an appendix with algorithm details.
>           We also added the source code and checkpoints for easy reproduction.
>
> "I think it's fair to call your approach evolutionary but you might want to discuss its relationship to beam search and in this scope discuss [A]."
>
> Response: [A] is a single objective method that evolves a single architecture from a simple to a more complex one.
>           It is a very different evolutionary approach than ours. We work with a population of individuals (architectures), then breed and mutate them to create new generations of architectures better than their parents.
>           It is the interactions between the individuals that is important in our method.
>
>
> "The comparison in Table 2 is not fair. You use the swift activation function and do not report the corresponding numbers for MobileNet or Mobile NASNet. Ramachandran et al. (2017) report these (75% and 74.2% for NASNet and MobileNet).
> Comparing the Dvolver architecture with ReLU activations to MobileNet does not indicate any improvements."
>
> Response: First we compare NASNet and DVOLVER without swish to prove that single objective search is less efficient than multi-objective method.
>           Then, in table 3, We want to push our cells to the maximum and compare with the best results found in previous works. We added NASNet swish and showed that DVOLVER still has better accuracy than NASNet.
>           We added the best results found in state-of-the-art works for mobile architecture and found that DVOLVER is very competitive.
>           Ramachandran et al. 2017(https://arxiv.org/pdf/1710.05941.pdf), state that MobileNet can benefit from swish and the publication date implies that it is MobileNet V1.
>           Their paper states that MobileNet with RELU is at 72% (Table 8) but we could not find any reference for MobileNet V1 (https://arxiv.org/abs/1704.04861) with that accuracy. Without more details, we cannot add this values in our paper but will if we can find more details.
>           We think it is fair to compare with the best results found in previous works and it is not the subject of our paper to optimize other architectures.
>
>
> "You mention that most previous approaches are only keeping track of the best solution while you evolve over a population. Maybe this sentence is not well written and something else is meant but now this statement is wrong."
>
> Response: We remove that statement, thanks for the catch.

---

### Official Review · AnonReviewer3 · 2018-11-02
**The results are competitive, but not too much innovation**

**Rating:** 5
**Confidence:** 3

**Review:**

The paper is easy to read. The authors did a job in describing the problem, concepts, and the proposed multi-objective optimization method. The computational results are on par with NASNet-A mobile.

It is good to know that we can use standard multi-objective method for neural architecture search. The implementation seems to be straightforward. The paper mainly uses existing ideas, but with some incremental improvements. It lacks novelty.

The time reduction of this method on ImageNet comes from transfer learning by training on CIFAR-10 first. As the paper admits this is not going to generalizing well. How good the method is if just using a single dateset? For CIFAR-10, is this method comparable with ENAS(https://arxiv.org/pdf/1802.03268.pdf)?

---

> ### Author Response · Authors · 2018-11-17
> **Review answers and extended experimental results**
>
> Thanks for your time and comments. Below we pull quotes from the review followed by responses.
>
> " It lacks novelty"
>
> Response: Our paper proves that multi-objective search can be more efficient than single objective method with extra value relevant for mobile neural network architecture design.
>           To our knowledge, no previous work has proved it.
>
>
> "As the paper admits this is not going to generalizing well. How good the method is if just using a single dateset?"
>
> Response: The main goal is to prove multi-objective search efficiency and for fast iteration, it is faster to perform the search on CIFAR-10 but it is not a requierement.
>           * We did the search on CIFAR-10 and then fully train on CIFAR-10 see Table 1.
>           * Searching on ImageNet can be done with our method but it is significantly more expensive. One of the future leads is to downscale Imagenet to make the search faster.
>
>
> "For CIFAR-10, is this method comparable with ENAS"
>
> Response: ENAS uses a clever approach to reuse weights and eliminate the need to retrain from scratch.
>           ENAS is a method to improve search efficiency. Our work is more centered on multi objective search. One of our future leads, is to use weight sharing with multi-objective search.

---

### Official Review · AnonReviewer1 · 2018-11-02
**Good results but limited novelty. Experimental comparison could be improved.**

**Rating:** 4
**Confidence:** 4

**Review:**

The paper proposes a multi-objective search algorithm that designs resource-efficient convolutional architectures. The key idea is to maintain a population of networks and to iteratively approach the Pareto front through evolution. The normal & reduction cells are searched on CIFAR-10 and then transferred to ImageNet. The resulting architectures empirically lead to better trade-offs than other baselines.

Pros:
The paper is well-written and easy to comprehend.
Results are competitive against strong baselines such as NASNet.
Resource budgets are handled in a principled manner with multi-objective optimization.

Cons:

My main concerns are on the technical novelty and experimental comparison.

Technical novelty:

The proposed algorithm seems highly similar to the existing multi-objective NAS algorithms, especially the ones based on Pareto optimality [1,2,3]. In Sect 2, the authors state that the main difference from prior works such as [2] and [3] is the usage of a different and larger search space and large-scale experiments. However, both aspects are of limited technical novelty.

Experimental comparison:

In sect 3.3, the authors say “we noticed that the original NASNet search space can greatly benefit from extra connections from any given block”. If the proposed algorithm was investigated in an enhanced version of the NASNet space, it would be unclear whether we should attribute the reported performance to the proposed multi-objective evolution or this additional search space engineering. It would be better to report the results using the original space as well for fair comparison.

The main claimed contribution is a multi-objective evolutionary algorithm. To demonstrate its effectiveness, it would be necessary to compare against existent multi-objective NAS strategies in the literature. Most of those strategies (e.g., scalarization, weighted product method) should be straightforward to implement on top of the current search space. The current results are less convincing since the authors only compared their method against single-objective baselines (e.g. NASNet, PNAS, AmoebaNet) which are completely unaware of additional dimensions of the desired objectives.

The networks are searched on CIFAR-10 and then transferred to ImageNet. Unlike most prior works (including the ones focusing on resource-constrained NAS), the authors did not the final performance of their architecture on CIFAR-10. It would be informative to report the CIFAR-10 results as well.

Other suggestions & questions:
The authors did not report their training setup for ImageNet. It would be good to include those details to ensure the readers are informed should there are any additional augmentations.

“uniform mutation and a crossover probability of 0.1” (sect 4.1)
It would be better to included more details on these evolution forces for reproducibility. These are also important component of the proposed algorithm.

“We manually select 3 architectures that we will be fully train on ImageNet in Section 4.2” (sect 4.1)
I believe this part needs more clarifications since there can be a large number of architectures on the Pareto front. What’s the criteria for manual selection?

[1] Elsken, Thomas, Jan Hendrik Metzen, and Frank Hutter. "Multi-objective architecture search for cnns." arXiv preprint arXiv:1804.09081 (2018).
[2] Kim Ye-Hoon, Reddy Bhargava, Yun Sojung, and Seo Chanwon. NEMO: Neuro-Evolution with Multiobjective Optimization of Deep Neural Network for Speed and Accuracy. ICML’17 AutoML Workshop, 2017.
[3] Dong, Jin-Dong, et al. "DPP-Net: Device-aware Progressive Search for Pareto-optimal Neural Architectures." arXiv preprint arXiv:1806.08198 (2018).

---

> ### Author Response · Authors · 2018-11-17
> **Experimental update and review answer**
>
> Thanks for your time and comments. Below we pull quotes from the review followed by responses.
>
> "The proposed algorithm seems highly similar to the existing multi-objective NAS algorithms, especially […] However, both aspects are of limited technical novelty."
>
> Response: We argue that to prove that a search technique is efficient it has to be done on a large search space where exhaustive and random search are not tractable.
>           Moreover, Our paper prove that on a given large search space, multi-objective search can be more efficient than single objective search with extra value when designing neural architecture for mobile device.
>           To our knowledge, no previous work has stated that point because they do not search in a search space already studied with a single objective method.
>
>           Our comments on the given references:
>           * [1]: build networks incrementally with a limited set of possible operations. In theory, their search space is infinite but in practice it is quiet small.
>                  objective functions are arbitrary separated into 2 categories: expensive and cheap. We show that it is not necessary.
>           * [2]: Their method initializes the search process from a given baseline network which is a strong prior and only explore the vicinity of already defined network instead of exploring the whole search space.
>                  They do not study transferability to ImageNet.
>           * [3]: We argue that their search space is too small (See Table 4 of our paper). It is centered around efficient handcrafted architectures. For 4 layers, there are only 324 possible cell architectures of which they evaluate around 200.
>                  With small increase in computation, it is possible to perform exhaustive search.
>
>
> " If the proposed algorithm was investigated […] to the proposed multi-objective evolution or this additional search space engineering."
>
> Response: Thanks for pointing the ambiguity, we fixed the paper with more explanations: NASNet, AmoebaNet and PNAS all have additional connections in the provided checkpoints and codes by the authors. They are not described in their respective papers.
>           We did an experiment with no extra connections and compared with NASNet without additional connections and found that Dvolver outperforms NASNet.
>
>
> "The main claimed contribution is a multi-objective evolutionary algorithm. To demonstrate its effectiveness, it would be […]  completely unaware of additional dimensions of the desired objectives.
> "
>
> Response: Our main goal is to compare single objective vs multi-objective search for neural network but we think your are right to ask for broader comparison and we updated the paper with results from DPP-Net (only paper with results on ImageNet).
>           We also made it clearer that when we compare with single objective method, only accuracy is to be considered.
>           One of our conclusion is that multi-objective search is still relevant even when looking for accuracy only.
>
> "It would be informative to report the CIFAR-10 results as well."
>
> Response: Again, thank you for pointing that out. We updated the paper with results for CIFAR-10.
>
>
> "The authors did not report their training setup for ImageNet. It would be good to include those details to ensure the readers are informed should there are any additional augmentations."
>
> and
>
> "It would be better to included more details on these evolution forces for reproducibility. These are also important component of the proposed algorithm"
>
> Response: We added more details in the paper and we provide the full source code for search and train on CIFAR-10 and also the code for train and evaluation on ImageNet (GPU and TPU) with the final checkpoints for all networks we presented on ImageNet.
>           We hope it will be enough to clear the shadow details in our setup.
>
>
> "What’s the criteria for manual selection?"
>
> Response: In general, interpreting the Pareto front is context sensitive. Ideally, we should train all the architectures in the Pareto front but it is very expensive.
>           Our selection process is as follow:
>           * take the cell with the best accuracy (DVOLVER-A), then take 2 more on Pareto front knees (where small drop in accuracy leads to great improvement in speed): DVOLVER-B & C.
>           * Then for all N, F and cells compute MACs and we select the few with MACs comparable with existing networks with want to compare with.
>
>           Our goal is to easy comparison with competitive architectures.

---

### Meta-Review · Area_Chair1 · 2018-12-13
**limited novelty**

**Confidence:** 5
**Recommendation:** Reject

**Metareview:**

The paper describes an architecture search method which optimises multiple objectives using a genetic algorithm. All reviewers agree on rejection due to limited novelty compared to the prior art; while the results are solid, they are not ground-breaking to justify acceptance based on results alone.